# An artificial triazole backbone linkage provides a split-and-click strategy to bioactive chemically modified CRISPR sgRNA

Lapatrada Taemaitree[1], Arun Shivalingam[1], Afaf H. El-Sagheer[1,2] & Tom Brown [1]

As the applications of CRISPR-Cas9 technology diversify and spread beyond the laboratory to diagnostic and therapeutic use, the demands of gRNA synthesis have increased and access to tailored gRNAs is now restrictive. Enzymatic routes are time-consuming, difficult to scale-up and suffer from polymerase-bias while existing chemical routes are inefficient. Here, we describe a split-and-click convergent chemical route to individual or pools of sgRNAs. The synthetic burden is reduced by splitting the sgRNA into a variable DNA/genome-targeting 20-mer, produced on-demand and in high purity, and a fixed Cas9-binding chemically-modified 79-mer, produced cost-effectively on large-scale, a strategy that provides access to site-specific modifications that enhance sgRNA activity and in vivo stability. Click ligation of the two components generates an artificial triazole linkage that is tolerated in functionally critical regions of the sgRNA and allows efficient DNA cleavage in vitro as well as gene-editing in cells with no unexpected off-target effects.

[1] Department of Chemistry, University of Oxford, Chemistry Research Laboratory, 12 Mansfield Road, Oxford OX1 3TA, UK. [2] Chemistry Branch, Department of Science and Mathematics, Faculty of Petroleum and Mining Engineering, Suez University, Suez 43721, Egypt. These authors contributed equally: Lapatrada Taemaitree, Arun Shivalingam.  Correspondence and requests for materials should be addressed to T.B. (email: tom.brown@chem.ox.ac.uk)

CRISPR-Cas9 genome editing has transformed our ability to manipulate genomes at the single-nucleotide level. The system is composed of a single-guide (sg) RNA that programmes a nuclease (Cas9) to cleave genomic DNA sequence specifically[1]. The resulting double-stranded breaks are recognised by the cell and repaired imperfectly, thus enabling the function of the cleaved sequence to be determined[2,3]. By partially inactivating the nuclease activity of Cas9 or creating dead Cas9 (dCas9) fusion proteins, it is even possible to image genomic loci in live cells[4], reprogramme the transcriptome[5,6], and create point mutated genomes[7,8]. At the core of these innovative applications, and a reason for CRISPR's far greater adoption than zinc-finger nuclease and TALEN systems, is the fact that the (d)Cas9 protein is guided to its target by a sgRNA that is designed using simple Watson–Crick base-pairing rules.

As the questions posed by researchers using CRISPR become more complex, the number of sgRNAs required has substantially increased. For example, high-content screens examining viral infection[9], profiling single-cell phenotypes[10] and studying epigenetic regulation[11] have used ~4500, ~2300 and ~450 arrayed sgRNAs, respectively, and many applications are likely being hindered by limited access to sgRNAs[12–14]. Enzymatic methods for the preparation of sgRNAs can be complex and time-consuming, and in the case of viral plasmid delivery, raise safety concerns. Methods for direct chemical synthesis of sgRNAs are therefore important; they can provide access to chemical modifications that enhance sgRNA stability[15–21] and reduce off-target effects[15,16,22,23]. However, 100-mer sgRNAs remain at the limit of solid-phase synthesis and the cost of oligoribonucleotides is far higher than deoxy variants, significantly increasing the barrier to their use. Efforts have been made to address these problems by using a bimolecular guide RNA system (a DNA-targeting ~42-mer crRNA that hybridises to a fixed ~80-mer tracrRNA) and incorporating 2′-F, 2′-OMe or deoxyribonucleotides into the crRNA/tracrRNA components, but this has come at the cost of larger constructs compared to the sgRNA design[20,23,24].

Here we synergise and refine these approaches, and use chemical ligation to create a simple method for preparing individual or pools of sgRNAs. Importantly, we demonstrate that a genomic DNA-targeting RNA bearing an alkyne, prepared on demand and in high purity, can be efficiently ligated to an invariant Cas9-binding RNA bearing an azide, made cost-effectively on a large-scale, by simple untemplated copper-catalysed azide-alkyne cycloaddition (CuAAC) chemistry. The resultant sgRNA contains an artificial triazole backbone at the point of ligation that enables effective Cas9-mediated DNA cleavage in vitro and in cells, with a comparable off-target profile to in vitro transcribed sgRNA.

## Results

### The scope of click chemistry in sgRNA construction

In our initial synthetic design, we split the sgRNA at the tetraloop of the repeat–anti-repeat hairpin to yield a truncated form of Nature's crRNA–tracrRNA system. It was envisaged that hybridisation of the two components ('self-templation') should facilitate CuAAC[25,26] chemical ligation, a reaction that has been reported to work well for RNA–RNA ligation[27–29]. A 37-mer crRNA was synthesised on solid phase with a terminal 3′-O-propargyl nucleotide introduced via a commercially available solid support, and a 66-mer tracrRNA was prepared with a terminal 5′-amino group that is post-synthetically labelled using a C6-azide NHS ester. The oligonucleotides were then chemically ligated using the CuAAC reaction, which after optimisation gave very efficient conversion to the clicked sgRNA (construct 2; Fig. 1a, b); the denaturing conditions (50% dimethyl sulfoxide (DMSO))[30] used remove the need for a ligation template or self-templated design

and simplify the system. Concerned by potential copper-induced artefacts that are undetectable by ultra performance liquid chromatography-mass spectrometry (UPLC-MS) but might affect CRISPR activity, we produced an equivalent system that cannot be contaminated with copper. To achieve this the 37-mer crRNA was prepared with a terminal 3′-amino group, labelled with DBCO-NHS ester and reacted with the azido-tracrRNA under copper-free strain-promoted azide-alkyne cycloaddition (SPAAC) conditions (construct 4; Fig. 1a, b). Pleasingly, both clicked sgRNAs enabled Cas9-mediated DNA cleavage in vitro at comparable levels to in vitro transcribed (IVT) sgRNA (constructs 2 and 4; Fig. 1c). These results are consistent with reports that Cas9 does not interact with the artificial tetraloop of sgRNAs allowing this position to accommodate large RNA extensions[31] or synthetic linkers[32] while remaining functional in cells.

Encouraged by this, the next goal was to reduce size and length of the click linker with the eventual aim of moving the ligation site into regions that form direct interactions with Cas9. An artificial seven-bond triazole backbone (Tz2; Fig. 1a) was selected due to its excellent biocompatibility with polymerases in vitro and in vivo[33–35]. To access this linkage, the 5′-hydroxyl group of unmodified TBDMS-synthesised tracrRNA was converted to an azide via an iodo intermediate on the solid support prior to oligonucleotide deprotection[36]. Subsequent deprotection and ligation of the 5′-azide tracrRNA with the 3′-O-propargyl crRNA proceeded efficiently (construct 1; Fig. 1a, b) and gave a Tz2-containing sgRNA that, promisingly, enabled Cas9-mediated DNA cleavage in vitro at levels comparable to IVT sgRNA (construct 1; Fig. 1c). Once more, a DBCO-labelled crRNA was ligated to the 5′-azide converted tracrRNA using the SPAAC reaction to circumvent potential copper-induced artefacts (construct 3; Fig. 1a, b), of which none were observed in the in vitro DNA cleavage assay (construct 3; Fig. 1c). To validate the importance of chemical ligation, Cas9-mediated DNA cleavage was performed as a function of gRNA concentration for the Tz2-containing sgRNA and its unligated starting materials (Fig. 1d). As anticipated, click ligation to reduce the number of components from a trimolecular (crRNA/tracrRNA/Cas9) to a bimolecular (sgRNA/Cas9) system eliminated undesirable RNA concentration-dependent effects on DNA cleavage which could be particularly problematic in vivo.

### An optimised split-and-click approach to sgRNAs

Chemical ligation of the crRNA and tracrRNA is an effective solution but it is still not perfect; over half of the crRNA sequence is invariant and, ideally, it should not be necessary to synthesise this part each time a new sgRNA is needed. Therefore, a more radical split of the sgRNA was explored – a ~20-mer that specifies the DNA target and a 79-mer that binds to Cas9. Based on X-ray crystallographic data[37], the ligation point was placed one base downstream of the DNA-targeting sequence between bases G1 and U2 (Fig. 2b). At this position, the artificial linkage should form only minor contacts with Cas9, and synthetically a terminal uracil base gives near-quantitative azide conversion (cf. lower yields for terminal purine bases)[36]. The 5′-azide modified 79-mer is significantly longer than the previously synthesised 66-mer tracrRNA reducing its yield (Supplementary Table 2) and potentially limiting its purity. However, the 79-mer was obtained at an acceptable level of purity using reversed-phase high-performance liquid chromatography (RP-HPLC) (Supplementary Fig. 1B). Any minor impurities that might escape detection by UPLC-MS are likely to have a negligible impact on Cas9 target specificity as the 79-mer lacks the DNA-targeting element. To improve the yield, the 79-mer was also synthesised with chimeric ribo/deoxyribonucleotides or 2′-OMe/ribonucleotides the

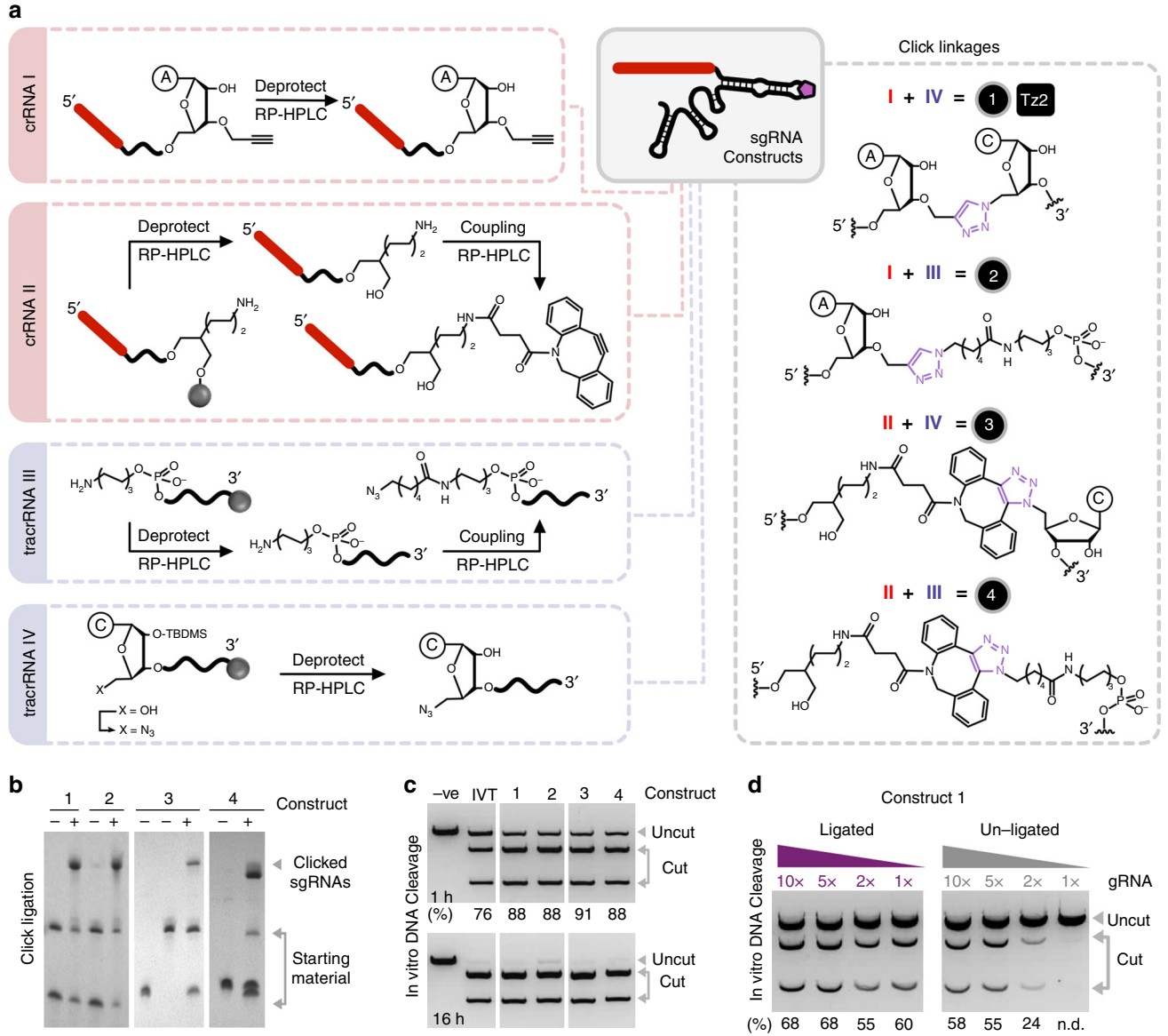

**Fig. 1** Synthesis and in vitro DNA cleavage activity of clicked crRNA–tracrRNA constructs. **a** illustrates the steps involved in oligonucleotide synthesis and the chemical ligation of the various oligonucleotide precursors to give sgRNA constructs containing artificial linkages 1–4 (full sequences are in Supplementary Table 1). Note that 5′-OH to 5′-azide conversion occurs on resin prior to deprotection. Only one of the two regioisomers of SPAAC linkages 3 and 4 is shown. **b** demonstrates the efficiency of CuAAC and SPAAC chemical ligation in generating constructs 1–4. '+' and '–' lanes indicate whether click ligation was performed or not respectively. Note that for construct 4 (but not construct 3), the click positive ('+') lane sample was heated during ligation, resulting in partial decomposition (hydration) of the DBCO starting material which consequently appears as two bands at the bottom of the gel. **c** shows the activity of constructs 1–4 in facilitating Cas9-mediated DNA cleavage relative to IVT sgRNA at 1 and 16 h time points. **d** shows the concentration-dependence of ligated and unligated construct 1 on in vitro DNA cleavage activity. The gRNA ratio was varied from 1 to 10 equivalents relative to Cas9 protein (30 nM). Cleavage values (under the gels) for **c** and **d** were quantified using the equation $f_{cut}/f_{total} \times 100$ and ImageJ, where $f$ stands for fraction. n.d. = not determined. Source Data are provided as a Source Data file

placement of which was based on previous reports[20,24]. DNA and 2′-OMe-RNA monomers assemble more efficiently than RNA monomers during solid-phase synthesis (Supplementary Table 2) and are less expensive. The DNA-targeting 3′-O-propargyl modified ~20-mer, on the other hand, is the ideal length for stringent RP-HPLC purification (Supplementary Fig. 1A), thereby minimising chemical artefacts that could give rise to off-target DNA cleavage activity, a problem that becomes harder to address as the oligonucleotide size grows (e.g. fully synthetic ~40-mer crRNAs and ~100-mer sgRNAs). The ~20-mers were also synthesised with and without a 5′-aminohexyl handle for potential post-synthetic functionalisation. CuAAC ligation of the various

alkyne and azide oligonucleotides gave good conversion to the clicked ~99-mer sgRNAs, demonstrating untemplated chemical ligation of two fully synthetic RNA oligonucleotides is efficient. These constructs were purified by denaturing PAGE (Supplementary Figs. 1C and 2, full sequences in Supplementary Data 1).

These constructs were then used in an in vitro Cas9-mediated DNA cleavage assay (Fig. 2c). Despite placing the Tz2 linkage very close to the seed-region, good activity was observed for the clicked sgRNA (50–54% DNA cleavage cf. IVT sgRNA in 1 h) with 5′-C6-amino containing constructs slightly outperforming the 5′-hydroxyl variant. More surprisingly the introduction of sugar modifications improved the activity of clicked sgRNA (13%

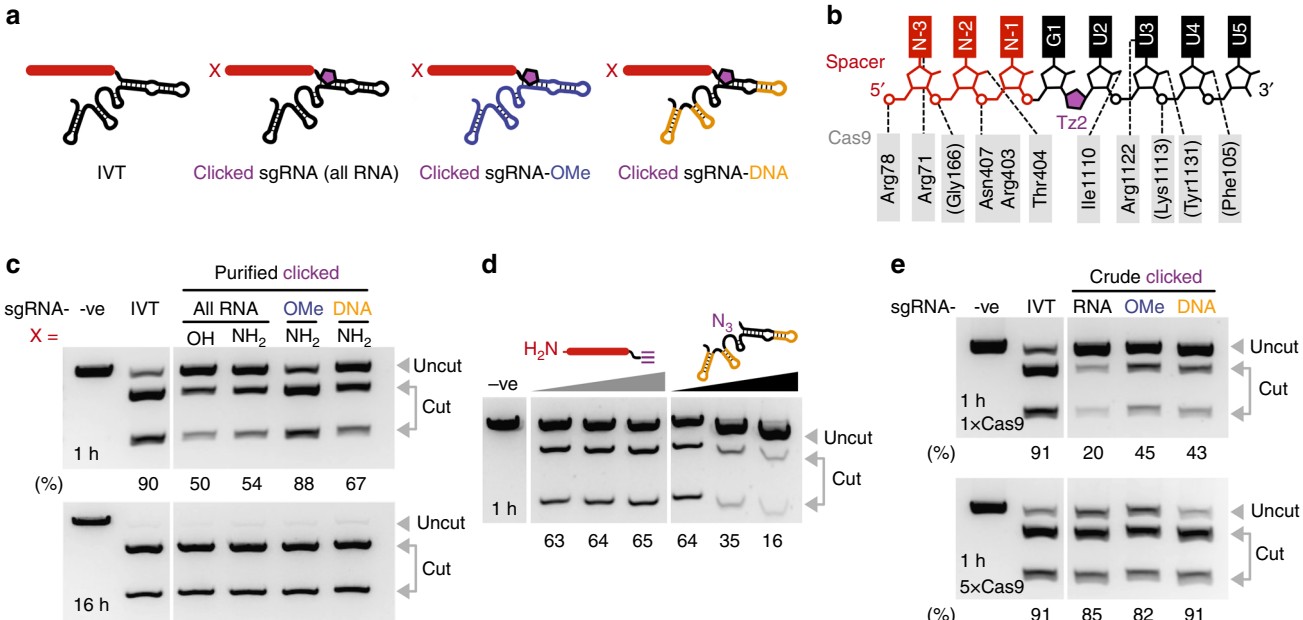

**Fig. 2** Clicked sgRNA constructs and their in vitro DNA cleavage activity. **a** illustrates the constructs tested (full sequences targeting plasmid pBR322 site 1 are in Supplementary Data 1). Note that when X = NH$_2$, the modification is a 5′-C6-NH$_2$ linker, and that when X = OH, there is no 5′ modification, only a 5′-OH group. **b** details the intermolecular interactions between Cas9 and the sgRNA based on PDB: 4OO8 (ref. [37]), as well as the position of the triazole linkage. Note that the spacer is red, the Cas9-binding RNA is black and the Tz2 linkage is purple. **c** shows the activity of these constructs in Cas9-mediated DNA cleavage relative to in vitro transcribed (IVT) sgRNA at 1 and 16 h time points using 1× Cas9. **d** shows the inhibitory effects of the 79-mer but not the ~20-mer on Cas9 activity (0, 2 and 4 equivalents relative to clicked sgRNA-DNA, sequences can be found in Supplementary Data 2). **e** demonstrates that desalting CuAAC clicked constructs, without additional PAGE purification, still enables DNA cleavage, and that increasing Cas9 concentration five-fold (from 30 to 150 nM) allows comparable cleavage to IVT sgRNA. Note that for these clicked constructs X = 5′-C6-NH$_2$. Cleavage values (under the gels) for **c** and **e** were quantified using an Agilent Bioanalyzer (Supplementary Fig. 4) where cleavage (%) = $f_{cut}/f_{total} \times 100$ and $f$ stands for fraction. Cleavage values (under the gel) for **d** was quantified using the same equation and ImageJ. Source Data are provided as a Source Data file

and 34% greater cleavage with deoxyribonucleotides and 2′-OMe modifications respectively in place of ribose, 1 h) and brought it to levels comparable to IVT sgRNA. This may be due to synergistic changes in sgRNA folding that offset the effects of the Tz2 linkage (e.g. stability of the sgRNA repeat–anti-repeat hairpin in which Tz2 is located). Importantly, for all constructs, near quantitative cleavage was observed at longer time points (16 h). Finally, a ~20-mer RNA strand with 5′-C6-NH$_2$ and 3′-serinol-alkyne was prepared and ligated to the 5′-azide 79-mer RNA to give an sgRNA with a very long linker (Supplementary Fig. 3). The presence of the extended linkage significantly impaired Cas9-mediated cleavage of DNA relative to the equivalent construct containing a Tz2 linkage (Supplementary Fig. 3) demonstrating the importance of the biocompatible Tz2 linkage in our design.

Clicked constructs were purified by denaturing PAGE. This level of purification is highly desirable for conventionally synthesised ~100-mer sgRNAs made on solid phase because, for such large constructs, phosphoramidite coupling efficiency decreases with oligonucleotide length, and the integrity and purity of the last 20 bases (5′-section) is essential to control DNA target specificity. However, such purification may not be needed for clicked sgRNAs as both the ~20-mer and 79-mer RNAs are already purified by RP-HPLC prior to ligation. In order to explore this option to the extreme, the effect of residual starting material on Cas9 activity was first evaluated. Taking a purified clicked sgRNA-DNA construct and titrating in either the 79-mer tracrRNA-DNA or the ~20-mer RNA (0, 2, 4 equivalents) confirmed that the Cas9-binding azido-79-mer inhibits target DNA cleavage but not the alkynyl-20-mer (Fig. 2d). Consequently, when preparing sgRNAs, the alkynyl-20-mer was used in slight excess, and after the CuAAC reaction, the sgRNAs were

simply desalted. Assuming near quantitative ligation efficiency, the crude sgRNA constructs were used in the in vitro DNA cleavage assay where they displayed reasonable activity (1 h, 1× Cas9; Fig. 2e). IVT sgRNA subject to the same click ligation and PAGE/desalting purification steps showed no change in DNA cleavage (Supplementary Fig. 5), suggesting changes in activity of the crude constructs in vitro relative to the purified constructs (1 h, 1× Cas9; Fig. 2c) is linked to residual 79-mer RNA and the difficulty in determining crude clicked sgRNA concentration. This was confirmed by increasing the Cas9 protein concentration to give a 1:1 ratio of protein to RNA, which pleasingly gave DNA cleavage at levels comparable to IVT sgRNA (1 h, 5× Cas9; Fig. 2e).

**Preparation of clicked sgRNA libraries.** One of the potential benefits of our chemical approach is the ease of mixing and matching DNA-targeting ~20-mers and reacting them with the invariant 79-mer to form pools of sgRNAs under denaturing conditions without fear of enzymatic bias that can occur during transcription. Therefore, to test this, CuAAC ligation was performed with a ~20-mer RNA containing either a 5′-C6-Cy3 or 5′-C6-ATTO 647N fluorescent dye. These fluorophores are well separated spectrally enabling quantification of CuAAC-mediated ligation via denaturing PAGE. Similar coupling efficiency was observed for both dye-labelled oligonucleotides (42% and 65% for Cy3 and ATTO 647N, respectively, note the dye-labelled 20-mers were used in a combined 1.5-fold excess to the 79-mer; Supplementary Fig. 6). Next, a set of sgRNAs were prepared as individual sgRNAs or as a pooled library by in vitro transcription (IVT) or click ligation (Fig. 3a). The individually prepared sgRNAs were

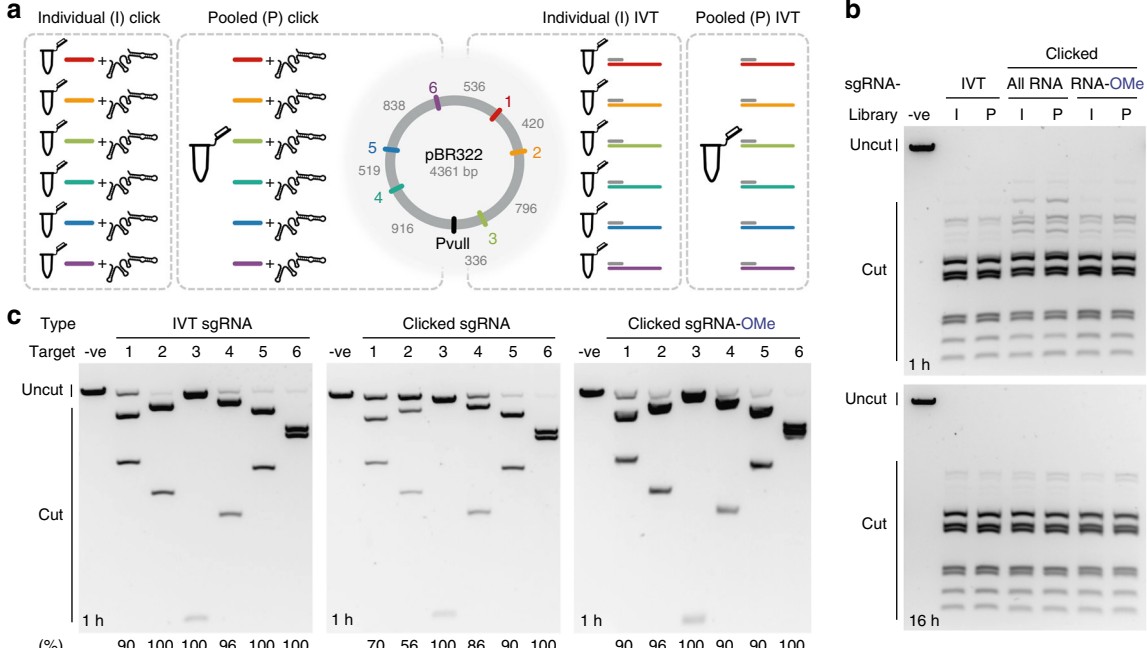

**Fig. 3** Clicked 20–79 sgRNA library activity in vitro. **a** illustrates preparation of sgRNAs by click ligation or in vitro transcription (IVT) for cutting a PvuII-linearised plasmid (colour-coded target sites). Each sgRNA was prepared individually (I) or in a pooled (P) one-pot reaction. Full sequences can be found in Supplementary Data 1. **b** shows the activity of the combined individually prepared sgRNAs (I) and pooled sgRNAs (P) in Cas9-mediated DNA cleavage at 1 and 16 h time points. Comparable DNA cleavage indicates minimal bias in library preparation. **c** shows the performance of individually prepared sgRNAs in DNA cleavage after a short time period (1 h). Clicked sgRNAs show comparable or lower DNA cleavage to IVT sgRNA but upon the introduction of 2′-OMe modifications activity is restored to IVT sgRNA levels. Cleavage values (under the gel) were quantified using an Agilent Bioanalyzer (Supplementary Fig. 7) where cleavage (%) $= f_{cut}/f_{total} \times 100$ and $f$ stands for fraction. Source Data are provided as a Source Data file

then mixed in equimolar quantities to provide a control for the pooled library preparations. Cutting using either the combined individual sgRNAs or the pooled sgRNAs gave comparable DNA cleavage patterns after 1 h and complete cleavage after 16 h (Fig. 3b). In combination, these results suggest there is minimal bias in pooled clicked sgRNA library composition. The simplicity of our approach allows the end user to perform the final single tube clicked sgRNA preparation step, enabling custom libraries to be readily made on demand.

The individual sgRNAs assayed at 1 h (Fig. 3c), where complete cleavage has not occurred (cf. 16 h; Fig. 3b, lanes 'I'), also offer a comparison of on-target activity. For IVT sgRNA and clicked sgRNA containing 2′-OMe modifications, excellent DNA cleavage ≥90% was observed for all sites. However, for the all-RNA clicked constructs, sites 1 and 2 gave lower cleavage. It is known that on-target activity is influenced by the target GC and position-specific nucleotide composition[38–41], but no distinct sequence–activity relationship could be identified for the all-RNA clicked constructs. We speculate that the differences are related to the Tz2 linkage influencing the global conformation of the target DNA:sgRNA duplex (A-/B-form), which is known to be recognised by Cas9 (ref. [42]) and has a mixed A-/B-character in hybrid duplexes[43]. Promisingly, the introduction of 2′-OMe modifications synergistically offsets this effect, possibly through changes in sgRNA folding stability (note that only the 2′-OMe modified 79-mer required a heated HPLC column for purification).

**Cellular activity and base pair specificity of clicked sgRNAs.** Evaluating the on-target activity of the clicked sgRNAs in live cells was a key priority. sgRNAs for the established *EMX1* target[44–46] were prepared bearing a 5′-C6-amino group with and without deoxy- or 2′-OMe ribonucleotides, and IVT sgRNA was

used as a control, a true gold standard for RNA integrity. Cells were transiently transfected with a Cas9-expressing plasmid and sgRNA, the genomic DNA was extracted and the target region amplified by PCR. Indel formation was then assessed using the T7E1 assay.

Clicked sgRNAs with all ribonucleotides were functional in cells (Fig. 4a). The activity of the modified all-RNA construct was lower than the control (16.6 ± 1.8% cf. 35.9 ± 1.2% for IVT sgRNA, s.e.m., $n = 6$, biological replicates); however, upon the introduction of site-specific 2′-OMe modifications indel formation was significantly improved (37.3 ± 2.5%, s.e.m., $n = 3$, biological replicates), and was comparable to IVT sgRNA. This is consistent with in vitro results (Fig. 2c) for shorter (1 h) rather than longer (16 h) time points. Conversely, incorporation of chimeric deoxyribonucleotides completely abolished gene editing (Fig. 4a) despite the good DNA cleavage in vitro, consistent with previous reports that this modification can significantly reduce activity in cellulo in the crRNA–tracrRNA system[24]. One possibility is that RNA–DNA hybridisation within the folded sgRNA-DNA could induce RNase H nuclease activity in cells, cleaving the RNA strand and thereby reducing the effective concentration of the construct. To test this, Cy3-labelled all-RNA or RNA-DNA clicked constructs were incubated with RNase H in vitro. The clicked sgRNA-DNA construct gave three distinct cleavage products whereas the all-RNA construct was stable to treatment supporting our hypothesis (Supplementary Fig. 8). Interestingly, the use of crude clicked sgRNA improved indels rates (23.9 ± 0.9%, s.e.m., $n = 3$, biological replicates) relative to the purified construct. The key difference in the crude preparation is that the excess ~20-mer RNA and unreacted 79-mer used during the click reaction are not removed. This may influence the lipid–RNA ratio in the transfection complex formed and hence transfection efficiency.

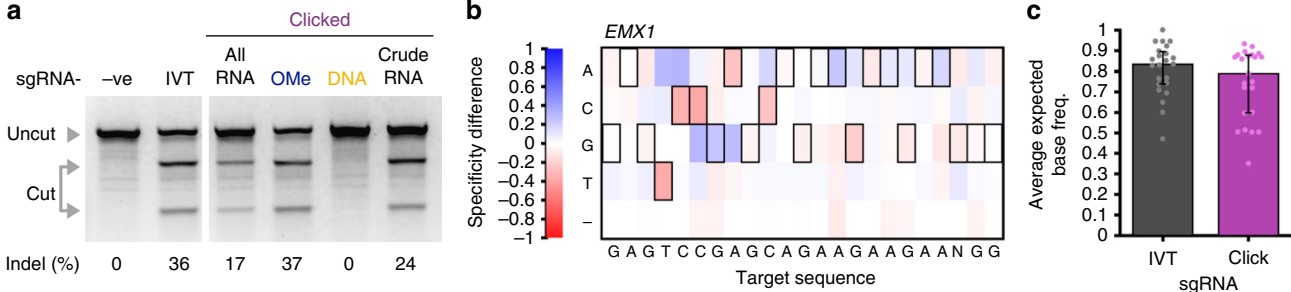

**Fig. 4** Clicked ~20–79 sgRNA construct activity in cells and their off-target profile. **a** is a representative gel demonstrating the ability of various sgRNA constructs to mediate indel formation in U2OS cells using the T7E1 assay. Cleavage was quantified using an Agilent Bioanalyzer (Supplementary Fig. 9, $n = 6$ for all-RNA clicked and IVT sgRNA and $n = 3$ for DNA and 2′-OMe modified clicked sgRNAs, biological replicates). *Indels* (%) = $(1-(1-f_{cut}/f_{total})^{0.5}) \times 100$, where $f$ stands for fraction. **b** illustrates the difference in specificity of IVT and clicked sgRNA as determined by CIRCLE-seq. All statistically significant cleavage sites (off- and on-target; Supplementary Fig. 10) were converted into a heat map of the frequency of each base observed at each position of the target sequence. Values for IVT sgRNA were subtracted from clicked sgRNA with the intensity of red indicating lower specificity and blue higher specificity of the specific base. Black boxes indicate the desired on-target base and '–' indicates deletion. **c** is a bar graph showing the median expected base frequency across the DNA target (including PAM). Error bars are the Q1 and Q3 quartiles ($n = 23$ positions of the target sequence) and dots are the individual points. Note that the clicked constructs have a 5′-C6-NH₂ modification. Full sequences for the oligonucleotide codes can be found in Supplementary Data 1. Source Data are provided as a Source Data file

Cautious of unintended changes in local base-pairing specificity and therefore potential off-target activity caused by the Tz2 linkage, we assayed for this possibility by comparing IVT sgRNA with the all-RNA clicked variant. The CIRCLE-seq protocol was chosen due to its NGS read-efficiency using genomic DNA, its sensitivity and its preference for over- rather than under-estimation of off-target effects[46]. Statistically enriched reads (Supplementary Fig. 10) were converted into a heat map of the observed bases at each position of the sgRNA and were compared to the expected base. To facilitate comparison, the observed frequencies for IVT sgRNA were subtracted from clicked sgRNA to give a specificity difference (Fig. 4b). In general values were close to zero (i.e. no difference in specificity between IVT and clicked sgRNAs) with a comparable number of positions that slightly favour or disfavour the desired on-target base. Taking the median on-target base frequency across the sgRNA indicates that there is no significant difference in specificity between clicked and IVT sgRNA (Fig. 4c, two-sided Wilcoxon ranksum test, $\alpha = 0.05$, $\rho = 0.29$, $n = 23$ positions of the target sequence).

In summary, we have demonstrated that the sgRNA can be fragmented, even at functionally critical regions, into smaller components, which after chemical ligation using the CuAAC reaction gives a sgRNA that performs efficient gene editing in cells comparable to IVT sgRNA. Our approach reduces the synthetic burden of sgRNA synthesis by allowing: (1) a fixed 79-mer component to be produced cost-effectively on a scale far higher than current enzymatic routes, with the potential for even larger-scale synthesis upon further development and (2) a highly pure variable DNA-targeting ~20-mer component to be produced on demand. Importantly the site-specific incorporation of modified nucleotides such as deoxy- or 2′-OMe ribonucleotides into our design is feasible and advantageous, as could be many other modifications such as bridged nucleic acids or 2′-O-methyl-3′-phosphonoacetates that enhance target specificity or backbone and other 2′-sugar modifications that improve sgRNA stability[15–23].

The successful use of the biocompatible triazole linkage, Tz2, allows radical fragmentation of the sgRNA. For example, the sgRNA could be split into multiple parts and combinatorially reassembled to access diverse sgRNA libraries containing chemical[24] or sequence modifications. Notably, the greatest strength of our split-and-click approach lies in screening numerous individual sgRNAs for function rather than preparing a single sgRNA. As the number of sgRNAs needed increases, the costs and time associated with repeated full-length sgRNA synthesis become greater and the merits of our approach become more evident. Moreover, clicked sgRNAs provide easier access to bespoke pools of modified sgRNAs (cf. full length sgRNA synthesis). Libraries of ~20-mer RNAs could be archived and custom libraries generated economically on demand with short turnaround times between sgRNA design and application.

## Methods

**RNA oligonucleotide synthesis**. RNA synthesis was performed on an Applied Biosystems 394 automated DNA/RNA synthesiser using a standard phosphoramidite cycle of detritylation, coupling, capping, and oxidation on a 1.0 μmole scale. Either nucleoside SynBase™ CPG 1000/110 (Link Technologies), 3′-O-propargyl G-ib 2′-lcaa CPG 1000 Å (Chemgenes) or 3′-amino-modifier C7 CPG 1000 Å (Link Technologies) were packed into a twist column (Glen research) for synthesis. 2′-O-TBDMS RNA phosphoramidites (A-tac, C-tac, G-tac and U where tac = *tert*-butylphenoxyacetyl; Sigma-Aldrich) were dissolved in anhydrous acetonitrile (0.1 M) immediately prior to use. Coupling, capping and oxidation reagents were 5-benzylthio-1*H*-tetrazole (0.3 M in acetonitrile; Link Technologies), fast deprotection Cap A (5% *tert*-butylphenoxyacetyl acetic anhydride in tetrahydrofuran)/Cap B (16% *N*-methylimidazole in tetrahydrofuran) and iodine (0.1 M in tetrahydrofuran, pyridine and water), respectively. The coupling time during RNA synthesis was 10 min. Stepwise coupling efficiencies were determined by automated trityl cation conductivity monitoring and in all cases were >97%.

**Solid-phase 5′-azide conversion of RNA**. After automated solid-phase synthesis, the fully protected resin-bound 5′-OH RNA was treated with methyltriphenoxyphosphonium iodide in anhydrous DMF (0.5 M, 1 mL) for 1 h at room temperature. The solid support was washed with dry DMF (3 × 1 mL) and dried with argon. A saturated sodium azide solution was then prepared by heating sodium azide (100 mg) resuspended in dry DMF (2 mL) for 10 min at 70 °C. After cooling to room temperature, the resin was treated with the solution for 5 h at 55 °C. The resin was then washed with DMF (3 × 1 mL), acetonitrile (3 × 1 mL) and dried with argon. The 5′-azido-RNA was cleaved from solid support and deprotected as described below.

**Solid-phase selective β-cyanoethyl removal**. Oligonucleotides bearing primary aliphatic amines were treated with diethylamine (20% in anhydrous acetonitrile) for 20 min at room temperature to suppress the formation of cyanoethyl adducts. The resin was then washed with acetonitrile (3 × 1 mL) and dried with argon.

**RNA deprotection**. The solid support was exposed to concentrated aqueous ammonia:ethanol (3:1 v/v) in a sealed vial for 2 h at 55 °C. The solution was filtered and the ammonia removed in vacuo. The ammonia-free solution was then freeze-dried, re-dissolved in a 1:1 mixture of dry DMSO (300 μL) and triethylamine trihydrofluoride (300 μL) and heated for 2.5 h at 65 °C. After cooling down to room temperature, sodium acetate (3 M pH 5.2, 50 μL) and butanol (3 mL) were added and the RNA was stored for 30 min at −80 °C. The RNA was then pelleted by

centrifugation ($12,000 \times g$, 30 min, 4 °C), the supernatant discarded and the pellet washed twice with 70% ethanol (750 μL). The pellet was then dried in vacuo, dissolved in water and desalted using a NAP™-10 column before further purification.

**Desalting purification**. Amicon Ultra Centrifugal Filters (Merck, UFC501096) were used according to the manufacturer's instructions. Typically, 3–5 washes were performed.

**RP-HPLC purification**. Oligonucleotides were purified using a Gilson HPLC system with ACE® C8 column (10 mm × 250 mm, pore size 100 Å, particle size 10 μm) with a gradient of buffer A (0.1 M TEAB, pH 7.5, where TEAB = triethylammonium bicarbonate) to buffer B (0.1 M TEAB, pH 7.5 containing 50% v/v acetonitrile) and a flow rate of 4 mL/min. For unmodified oligonucleotides (and also all 5′-azide converted oligonucleotides), the gradient was 20–30% buffer B over 21 min. For other oligonucleotides, the gradient was suitably adjusted. Note that for 2′-OMe-modified oligonucleotides, the column was heated to 55 °C to disrupt secondary structures.

**Non-templated CuAAC ligation**. 3′-Alkyne RNA (750 pmol in 1 μL $H_2O$) and 5′-azido-RNA (500 pmol in 1 μL $H_2O$) were mixed with $MgCl_2$ (100 mM, 0.5 μL), triethylammonium acetate buffer (2 M, pH 7, 1 μL), DMSO (5 μL) and fresh ascorbic acid (125 mM, 1 μL). While degassing the oligonucleotide solution with argon in a 0.5 mL eppendorf, a solution of $CuSO_4$-tris(3-hydroxypropyltriazolylmethyl)amine (250 mM in 55% v/v DMSO to $H_2O$, 0.5 μL) was added and the reaction (final volume = 10 μL) was left for 1–2 h at room temperature. The sample was then diluted with water and desalted using an Amicon Ultra Centrifugal Filter.

**Non-templated SPAAC ligation**. 3′-DBCO RNA (1250 pmol) and 5′-azido-RNA (500 pmol) were mixed in a NaCl solution (0.2 M, 7.9 μL) containing EDTA (0.5 M, 0.1 μL) and HEPES (0.1 M, pH 7.5, 2 μL) and were either left for 1–2 h at room temperature or heated for 5 min to 95 °C then cooled to 25 °C by 1 °C/min and left for 1–2 h at room temperature.

**Denaturing PAGE purification**. Oligonucleotides were mixed with formamide (50% v/v) and loaded onto a denaturing 8–10% polyacrylamide gel (1× TBE buffer containing 7 M urea, W × D × H = 18 × 0.2 × 24.4 cm) and separated at 20 W for 2–3 h. DNA bands were visualised under UV, excised, crushed, soaked in water (for DNA, ~15 mL) or buffer (for RNA, 50 mM Tris-HCl pH 7.5 containing 25 mM NaCl, ~15 mL) overnight at 37 °C with vigorous shaking. After gravity filtration to remove the gel, the oligonucleotide solutions were concentrated in vacuo and desalted using two consecutive NAP™-25 columns according to the manufacturer's instructions.

**Cell culture and transfection**. U2OS cells (kindly gifted by Prof. G. Dianov) were cultured in DMEM (high glucose, HEPES-buffered, no phenol red, with glutamine; Life Technologies, cat. no. 21063029) supplemented with fetal bovine serum (10% v/v, Life Technologies, cat. no. 10270098) in a humidified incubator at 37 °C with 5% $CO_2$. The cell line was not mycoplasma tested or authenticated.

$1.5 \times 10^5$ cells were seeded in six-well plates. After 48 h, the media was replaced with fresh media and the cells were transfected using Lipofectamine 3000 (3.75 μL), P3000 (10 μL) and pSpCas9(BB)-2A-Puro-v2-Broccoli (5 μg) according to the manufacturer's protocol. After 12 h, the media was replaced with fresh media and the cells were transfected using Lipofectamine RNAiMAX (7.5 μL) and the appropriate sgRNA (25 pmol, 2.5 μL) according to the manufacturer's protocol. After 12 h, the media was replaced with fresh media. Cells were then allowed to recover and grow for a further 72 h with media replaced when necessary.

**Reporting summary**. Further information on experimental design is available in the Nature Research Reporting Summary linked to this article.

## Data availability
Sequencing data that support the findings of this study have been deposited in the NCBI Sequencing Read Archive with the accession code PRJNA512007. The source data underlying Figs. 1B–D, 2C–E, 3B–C, 4A–C; Supplementary Figs. 1, 2, 3B–C, 4, 5, 6, 7, 8, 9, 10; Supplementary Tables 1, 3, 4, 5, 6 and Supplementary Data 1 and 2 are provided in the Source Data file. All data for gels, graphs and mass spectrometry are provided as a Source Data file.

## Code availability
Post-CIRCLE-seq data plotting code is available upon request.

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

## Acknowledgements

This work was supported UK BBSRC grants BB/J001694/2 (Extending the boundaries of nucleic acid chemistry), BB/M025624/1 (Next-generation DNA synthesis), and BB/R008655/1 (New and versatile chemical approaches for the synthesis of mRNA and tRNA). We also thank ATDBio Ltd for supporting A.H.E.-S. and A.S. as well as the Royal Thai Government for supporting L.T.

## Author contributions

L.T., A.S., A.H.E.-S and T.B. were involved in the design of the study and co-wrote the paper. A.H.E.-S. and L.T. prepared clicked sgRNA constructs. L.T. conducted in vitro assays and in cell assays. A.S. conducted in cell assays and performed sequencing.

## Additional information

**Competing interests:** A.H.E.-S. and T.B. are co-inventors on US patent 8,846,883 B2 'Oligonucleotide Ligation'. The remaining authors declare no competing interests.

