## [Peer Review File · Nature Communications]

Reviewers' comments:

Reviewer #1 (Remarks to the Author):

The authors provide a method to generate a sgRNA/DNA hybrid that is active in CRISPR/Cas9 assays. In this work they describe how either individual or pools of sgRNAs can be made rapidly through a convergent chemical ('click') ligation of a variable 20-mer RNA that targets the genome and a Cas9-binding 79-mer RNA-DNA chimeric but fixed sequence. The click ligated sequence is nearly as functionally active as an in vitro transcribed sgRNA. The significance of this method is that it allows more ready access to modified sgRNA for use in CRISPR/Cas9 assays for biological and biochemical analyses. Sequences of the length of sgRNA are at the limits of synthesis for practical purposes, so the strategy to split the variable and fixed regions is pertinent. Further, the use of a DNA/RNA chimera (previously shown to be viable) for the fixed region reduces the synthetic and cost barrier to the modified-sgRNA – though this works for in vitro CRISPR/Cas9 assays, but not for in vivo gene editing (which was known). The data is sound and presented in a logical manner and the manuscript is easy to follow.

This work will be of excellent value to a large community of chemical biologists and I do recommend publication Nat Communications.

I do have one specific concern and other minor comments as noted below

The authors describe different click linkages they tried (copper and copper-free variants) and their optimization of their construct - and they do take the step of showing that the DNA-RNA chimera is alright in in vitro assays but then not suitable in vivo (although that's not totally unexpected). However, given that the 79mer needed to be all RNA and not DNA/RNA which worked in vitro, I was wondering why the authors did not test a ligated 79mer. Perhaps a large amount of DNA is not tolerated in vivo, but then a single (or two residues say in the middle of a duplex) with a click linkage would be tolerated in vivo. Splitting the 79mer also into say 2 syntheses would significantly enhance the access. I'm not asking that the authors try a lot of optimization – but at least where the DNA is tolerated in vitro – try a click linkage (with RNA or even DNA flanking residues) and see the effect (in vitro and in vivo if promising).

Minor comments

Figs 1C, 2D and 2E should have cleavage % noted (as done for 2C) – (no need to write the term Cleavage (%) at the bottom of fig as done for 2C if space issues. Can simply put in the number and fig legend can note that numbers represent cleavage %. Alternately this can be included in SI – but seems like a lot of redundancy just for that number. However, the cleavage % is valuable quantitative information – otherwise it is just qualitative (and further obfuscated or subjective due to printing or screen contrast)

Pg 5 Choice of deoxy sites – was this just based on published sequence in ref 18? If so perhaps a slightly more specific statement than "To improve the yield, the 79-mer was also synthesised with chimeric ribo/deoxyribonucleotides 18" as this seems to be more about the yield (which is important). Should note that the choice of deoxy residues – amount and placement – was from 18 and not further optimized.

(so this would be analogous to the authors noting their ligation point was based on crystallographic data (ref 32) as on pg4)

Pg 5 increased 'activity' is alright to claim but should note that it increased % cleavage (not increase in % activity)

So: Surprisingly the introduction of deoxyribonucleotides very slightly increased sgRNA activity (~5% greater cleavage at 1h), and the 5'-C6-amino containing sgRNAs outperformed 5'-hydroxyl variants (~10% greater cleavage at 1h).

Pg 8 typography error: "chemically ligation" – should be "chemical ligation"

Reviewer #2 (Remarks to the Author):

Thank you for submitting the manuscript. However, I really can't recommend that it be published in Nature Communications. I have several comments about the major points of the manuscript and the rationale used to justify submission to a journal with broad readership.

Overall: I really don't find the work highly significant within the context of what has been previously published to date. It has been shown many times that "click chemistry" can be used to join two oligonucleotides. The position that was chosen, the stem-loop between the crRNA and tracrRNA is known to tolerate modifications and non-nucleotide linkages and the idea that a triazole linkage in that position would affect specificity would be unexpected. The appropriate place for this manuscript is a very focused nucleic acids chemistry journal such as Nucleosides Nucleotides and Nucleic Acids. However, if this manuscript is submitted to a more focused readership there needs to be much substance included in the manuscript.

Specific:

One of your central arguments is that 100mer RNAs are beyond the scope of routine chemical synthesis, that may have been true 10 or 15 years ago, but newer chemistries and engineering techniques have been developed that allow skilled researchers and oligo houses to provide these materials routinely. It has been shown that RNAs more than 200 nucleotides in length are achievable in high yields and high purity by chemical synthesis.

The work flow that you describe doesn't seem to support the idea that this will be a significant improvement in the ability to make and synthesize 100mer RNAs. If you want to make this argument please support it with data; make a 96 well-plate of sgRNAs and show the individual yields, purities, in a table and the time it took you to complete, then compare that to the chemical synthesis by one of the modern methods for RNA chemical synthesis.

You need to do more in depth characterization of your final RNA products. The deconvoluted mass spectrum you show in the supplemental material is of very low resolution. I would suggest that you find a LC separation method of significantly longer than 5 minutes and evaluate and report the complete characterization using extracted mass, this is now typical. It is especially important because you argue that your constructs could be used for therapeutic oligonucleotides. There are many publications on the characterization of therapeutic oligonucleotides by LC/MS.

In your supplemental material, you describe two methods for RNA synthesis, but we don't know which one was used in which experiment. You should see significantly different product profiles by those two chemistries and you only report trityl yields from the old ABI conductivity detector.

You need to revisit your use and choice of references, in many cases in-text citations don't seem to match what is stated, and your choice of references seem to be selective not based upon what was first published or inclusive. If you want to have a specific reference we should know why you chose that specific reference otherwise you should be inclusive: as an example Cromwell, C. R. et al. was far from being the first to publish the use of nucleotide modifications to increase specificity in CRISPR, why is it single out? This is especially concerning because LNA or LNA-like modifications to enhance specificity have not borne out to be widely reproducible as a result of increase in off-target binding just due to the increased melting temperature of partially homologous sequences; making short tight binding sequences with low sequence complexity results in binding to larger numbers of off-targets throughout the genome

Reviewer #3 (Remarks to the Author):

The manuscript entitled: "Click and play CRISPR sgRNA" is developing a split-and-click convergent chemical route to individual sgRNAs by splitting the sgRNA into a variable genome-targeting 20-mer, a fixed Cas9-binding 79-mer, and ligating of these two components using CuAAC reaction. With that, the study found that the sgRNA could be fragmented, even at functionally critical regions (one base downstream of the DNA-targeting sequence). Moreover, the DNA cleavage

activity of the “clicked” sgRNA with an artificial triazole linkage (Tz2) in vitro as well as in vivo was confirmed. This work provides a novel sgRNA synthesis approach with reduced synthetic burden comparing with existing chemical routes. Overall, I feel that this work is well executed, provides useful and important tool for further exploration. It is hence of broad interest and readership in my opinion.

In its present form, however, it has some shortcomings that need be corrected before the manuscript is accepted for publication. The following major issues need to be addressed first.

1. Quantitative data for the whole chemical synthesis is needed, especially for the yields when comparing different synthetic routes, and with or without chimeric ribo/deoxyribonucleotides.
2. The authors claimed that their methods could be applied for the synthesis of sgRNA pools, but they demonstrated very few examples of sgRNA sequences. Is there any sequence dependent effect of their method? How about the performance when synthesizing a large sgRNA pool?
3. To use “crude” sgRNA is a great idea, but they only showed the data for in vitro DNA cleavage, how about using crude sgRNA for the in vivo test?
4. Site-specific incorporation of modified nucleotides will be an important topic for this study. They have shown some interesting phenomenon when incorporating deoxyribonucleotides into the sgRNA, and they found that introduction of deoxyribonucleotides slightly increased sgRNA activity in the in vitro test, while the incorporation of chimeric deoxyribonucleotides completely destroyed gene editing activity in the cell. The authors should give more explanations on this point.

We thank the reviewers for their insightful comments, which prompted us to carry out a series of additional experiments that have strengthened the paper. As suggested by the editor, we have focused on the comments of reviewers 1 and 3 but have also addressed reviewer 2. Below is our detailed response to their main comments.

Reviewer 1. Given that the 79mer needed to be all RNA and not DNA/RNA which worked *in vitro*, I was wondering why the authors did not test a ligated 79mer. Perhaps a large amount of DNA is not tolerated *in vivo*, but then a single (or two residues say in the middle of a duplex) with a click linkage would be tolerated *in vivo*. Splitting the 79mer also into say 2 syntheses would significantly enhance the access. I'm not asking that the authors try a lot of optimization – but at least where the DNA is tolerated *in vitro* – try a click linkage (with RNA or even DNA flanking residues) and see the effect (*in vitro* and *in vivo* if promising).

Splitting the 79-mer in two components and carrying out DNA cleavage experiments, while feasible, would be a significant endeavour. Nevertheless, we agree with the sentiments of the reviewer that more chemical modification of the 79-mer would be interesting. To this end, we explored the site-specific incorporation of 2'-OMe modifications (which also couple better than RNA phosphoramidites and are still cheaper) based on recent reports in the crRNA/tracrRNA system (A. Mir *et al.*, *Nat. Commun.* 2641, 2018). This modification allowed significant improvements in 79-mer yield, and sgRNA activity *in vitro*. Most importantly, the 2'-OMe modified click constructs are active in cells with indel formation comparable to IVT sgRNA. These positive results have been highlighted throughout the manuscript with updates made to Figures 2, 3 and 4.

Reviewer 2. The position that was chosen, the stem-loop between the crRNA and tracrRNA is known to tolerate modifications and non-nucleotide linkages and the idea that a triazole linkage in that position would affect specificity would be unexpected.

We wish to re-iterate that the final design places the artificial triazole backbone close to the seed-region of the sgRNA (one base downstream). In this functionally critical region, the short Tz2 linkage is tolerated and allows segmentation of the sgRNA into a DNA-targeting 20-mer that can be highly purified, and a Cas9-binding 79-mer that can be prepared on larger scale. This allows far more flexibility in the scale of sgRNA preparation based on the demands of the end user.

Reviewer 2. One of your central arguments is that 100mer RNAs are beyond the scope of routine chemical synthesis, that may have been true 10 or 15 years ago, but newer chemistries and engineering techniques have been developed that allow skilled researchers and oligo houses to provide these materials routinely. It has been shown that RNAs more than 200 nucleotides in length are achievable in high yields and high purity by chemical synthesis.

The claim by reviewer 2 that 200-mer RNA constructs can be made chemically in high yield and high purity is incorrect. We have contacted the major 'oligo houses' Eurofins, Merck (Sigma-Aldrich), IDT Technologies and Bio-synthesis to check this. They offer RNA oligonucleotides up to 80, 45, 60 and 90-mer in length respectively as standard commercial services. Longer oligonucleotides can be supplied by special order: Merck offer maximum

length 100-mers for £2896, HPLC-purified, but they cannot guarantee a specific level of purity. IDT offer max 120-mers but only as unpurified constructs with no guarantee of quality. The most extreme claim we could find is from Dharmacon who offer maximum length 134-mer RNAs for £1300, HPLC-purified (1-2 nmoles) but can only guarantee 70% purity. The above costs and yields do not allow for any chemical modifications which can make synthesis more challenging and expensive. These significant costs and lack of guarantee of purity for long RNA oligonucleotides supports our assertion that the synthesis of long RNA is not routine, and better routes are in demand.

Reviewer 2. The work flow that you describe doesn't seem to support the idea that this will be a significant improvement in the ability to make and synthesize 100mer RNAs. If you want to make this argument please support it with data; make a 96 well-plate of sgRNAs and show the individual yields, purities, in a table the and time it took you to complete, then compare that to the chemical synthesis by one of the modern methods for RNA chemical synthesis.

The synthesis of high quality long (100-mer) RNA is not amenable to 96-well plate synthesis; otherwise the oligo houses would be supplying it routinely. In this study we carried out more precise and rigorous column-based RNA synthesis, which importantly is amenable to significant scale-up. Our new results in Supplementary Table 2 (see next page) demonstrate that shorter oligonucleotides can be prepared in higher yields than longer ones. The synthesis of 96 x 100-mer RNAs by a column-based method would therefore be an expensive undertaking that only confirms these observations. This length-dependent limitation on RNA synthesis is the reason why we have developed the modular methodology presented in this paper, i.e. the synthesis of a single invariant 79-mer and several variable gene-targeting 20-mers. Major efforts can be devoted to the synthesis and purification of the 79-mer, which is within the scope of current oligo houses commercial services. This 79-mer would then become a valuable resource for the synthesis of any desired sgRNA. Surely this is better than making a 100-mer RNA every time a new sgRNA is required.

Reviewer 2. The deconvoluted mass spectrum you show in the supplemental material is of very low resolution. I would suggest that you find a LC separation method of significantly longer than 5 minutes

The original traces were zoomed in to highlight the region of interest. We have now zoomed out to show the full run time. For resolution, we believe mass within 1-3 units of the expected molecular weight is typical for oligonucleotides.

Reviewer 2. In your supplemental material, you describe two methods for RNA synthesis, but we don't know which one was used in which experiment.

We have now clarified this.

Reviewer 2. You need to revisit your use and choice of references ... If you want to have a specific reference we should know why you chose that specific reference otherwise you should be inclusive.

We have tried to broaden the references to cover all chemical modifications of sgRNA.

Reviewer 3. Quantitative data for the whole chemical synthesis is needed, especially for the yields when comparing different synthetic routes, and with or without chimeric ribo/deoxyribonucleotides.

For quantitative data, we have chosen to focus on the final optimised 20–79 design and the constructs that proved functional in cells – namely the all-RNA and 2'-OMe modified 'clicked' sgRNAs. For reference we have also included yields for IVT sgRNA. We hope this demonstrates synthetic routes offer good scalability. This data is now provided in Supplementary Table 2 (below) and referred to in the 20–79 sgRNA preparation paragraph of the manuscript.

Supplementary Table 2. Representative yields for 'clicked' ~20-mer–79-mer sgRNAs and IVT sgRNA for cutting plasmid pBR322 at site 3. "Crude" refers to samples that are desalted after deprotection using a NAPTM-10 column to remove protecting groups that would otherwise interfere with concentration determination. Quantities were determined by dividing the optical density (OD) at 260 nm by the appropriate oligonucleotide molecular extinction coefficient. Note that 1 μ mol synthesis scale is typically ~30 mg resin. * = yield after HPLC. ** = yield after denaturing PAGE. *** = IVT sgRNA yield from 20 μ L reaction. The click constructs were formed from chemical ligation of the oligonucleotides listed in Supplementary Table 5 and the IVT sgRNA from transcription of the templates listed in Supplementary Table 6.

RNA	Purification	End	Resin (mg)	OD	Vol (mL)	pmol	pmol/mg	Yield* (%)
24-mer crRNA (CR3)	Crude	3'-alk	24.6	24.2	1.2	109420	4442	42.9
	HPLC	3'-alk	24.6	41.6	0.3	47023	1911	
79-mer tracrRNA (TR1)	Crude	5'-OH	2.5	5.05	1.5	8430	3372	10.9
	Crude	5'-N ₃	7.0	12.62	1.5	21066	3009	
	HPLC	5'-N ₃	7.0	3.44	0.6	2297	328	
79-mer 2'-OMe tracrRNA (TR3)	Crude	5'-OH	5.2	14.7	1.5	24538	4719	25.2
	Crude	5'-N ₃	14.5	42.7	1.5	71278	4916	
	HPLC	5'-N ₃	14.5	26.85	0.6	17928	1236	

Construct	24-mer	79-mer	Product	Product	Product	Yield**
-----------	--------	--------	---------	---------	---------	---------

	(pmol)	(pmol)	(OD)	(μ L)	(pmol)	(%)
Clicked sgRNA (Site 3)	375	250	0.70	72	43	17.2
Clicked sgRNA-OMe (Site 3)	750	500	0.71	90	71	14.2
IVT sgRNA (Site 3)***	-	-	8.58	50	369	-

Reviewer 3. The authors claimed that their methods could be applied for the synthesis of sgRNAs pools, but they demonstrated very few examples of sgRNA sequences. Is there any sequence dependent effect of their method? How about the performance when synthesising a large sgRNA pool?

To illustrate library-type applications, we prepared sgRNAs for six targets individually or as a pooled library by 'click' ligation. Combining the individually-prepared sgRNAs and comparing the activity to the pooled library shows minimal difference in DNA cleavage patterns and complete DNA cleavage at longer time points. Looking at the activity of the individual sgRNAs shows some differences in activity at after short period (1 h) for the all-RNA construct, which we tentatively suggest is due to triazole influencing the A-/B-form geometry of the DNA:RNA hybrid duplex recognised by Cas9, a global property of the desired sequence. Promisingly, the use of 2'-OMe modifications minimises this effect. These results are now reflected in two new sections of text and the new figure below.

Figure 3 | 'Clicked' 20–79 sgRNA library activity *in vitro*. **A** illustrates preparation of sgRNAs by 'click' ligation or *in vitro* transcription (IVT) for cutting a PvuII-linearised plasmid (colour-coded target sites). Each sgRNA was prepared individually (I) or in a pooled (P) one-pot reaction. Full sequences can be found in Supplementary Table 3. **B** shows the activity of the combined individually-prepared sgRNAs (I) and pooled sgRNAs (P) in Cas9-mediated DNA cleavage at 1 and 16 h time points. Comparable DNA cleavage indicates minimal bias in library preparation. **C** shows the performance of individually-prepared sgRNAs in DNA cleavage after a short time period (1 h). 'Clicked' sgRNAs show comparable or lower DNA cleavage to IVT sgRNA but upon the introduction of 2'-OMe modifications activity is restored to IVT sgRNA levels. Cleavage values (under the gel) were quantified using an Agilent Bioanalyzer (Supplementary Fig. 7) where cleavage (%) = $f_{\text{cut}}/f_{\text{total}} \times 100$.

Reviewer 3. To use "crude" sgRNA is a great idea, but they only showed the data for *in vitro* DNA cleavage, how about using crude sgRNA for the *in vivo* test?

We have now performed this experiment. The crude preparation works in cells and a reason for difference in activity compared to the purified preparation in cells is suggested –

"Interestingly, the use of crude 'clicked' sgRNA improved indels rates ($23.9 \pm 0.9\%$, s.e.m., $n = 3$, biological replicates) relative to the purified construct. The key difference in the 'crude' preparation is that the excess ~20-mer RNA and unreacted 79-mer used during

Reviewers' comments:

Reviewer #1 (Remarks to the Author):

The authors have addressed most, if not all, concerns about the initial manuscript in this revised version.

Additional work reported in incorporating chemically modified residues is useful.

This is not a perfect manuscript and the method described may or may not become the method of choice for screening CRISPR sgRNA constructs. Still this is a valiant effort from the authors that will inform studies combining and screening of CRISPR systems.

Even though DNA and RNA synthesis is routine - depending on the size/length of the sequence (particularly >100 residues) is at or beyond the limits of routine practice. Thus the authors efforts and report here to use click Chemistry to get long sequences of sgRNA in a combinatorial manner and test in a CRISPR screen is valuable for the community.

I recommend publication

Reviewer #2 (Remarks to the Author):

I have read the revised manuscript, reviewed the data and still my recommendation is that the manuscript should not be published in a journal with broad readership such as Nature Communications.

Even if the manuscript were to be considered for a more focused journal there are still many revisions that need to be made.

1. In my opinion "Click and play CRISPR sgRNA" is not the title of a scientific manuscript it is the title of a piece of marketing literature. What is "play"? I would prefer to see "The use of copper(I)-catalyzed azide-alkyne cycloaddition for the ligation of smaller RNAs into single guide RNAs" or even "The use of Click Chemistry for the ligation of smaller RNAs into single guide RNAs"? I will leave this to the judgement of the editors.

2. Once again, the thesis that 100mer RNAs are difficult to obtain in high yield and purity is just factually inaccurate or self-serving.

Reviewer 2. One of your central arguments is that 100mer RNAs are beyond the scope of routine chemical synthesis, that may have been true 10 or 15 years ago, but newer chemistries and engineering techniques have been developed that allow skilled researchers and oligo houses to provide these materials routinely. It has been shown that RNAs more than 200 nucleotides in length are achievable in high yields and high purity by chemical synthesis.

The claim by reviewer 2 that 200-mer RNA constructs can be made chemically in high yield and high purity is incorrect. We have contacted the major 'oligo houses' Eurofins, Merck (Sigma-Aldrich), IDT Technologies and Bio-synthesis to check this. They offer RNA oligonucleotides up to 80, 45, 60 and 90-mer in length respectively as standard commercial services. Longer oligonucleotides can be supplied by special order: Merck offer maximum length 100-mers for £2896, HPLC-purified, but they cannot guarantee a specific level of purity. IDT offer max 120-mers but only as unpurified constructs with no guarantee of quality. The most extreme claim we could find is from Dharmacon who offer maximum length 134-mer RNAs for £1300, HPLC-purified (1-2 nmoles) but can only guarantee 70% purity. The above costs and yields do not allow for any chemical modifications which can make synthesis more challenging and expensive. These significant costs and lack of guarantee of purity for long RNA oligonucleotides supports our assertion that the synthesis of long RNA is not routine, and better routes are in demand.

The authors have chosen to avoid the fact that 100mer RNAs, which is what they demonstrate in the manuscript, are now routine and rather to argue that 200mer RNAs are not routine to justify making 100mer RNAs by CuAAC ligation? 200mer RNAs are currently the state of the art and are not be widely available, but 100mer sgRNAs (which is the subject of their manuscript) are widely available. There are many companies that offer 100mer single guide RNA:

<https://www.synthego.com/products/crispr-kits/advanced-rna>

<https://www.trilinkbiotech.com/mRNA/longrna.asp>

<https://www.agilent.com/en/promotions/chemically-synthesized-crispr-guides>

<https://www.biolegio.com/products-services/synthetic-sgrna/>

https://www.chemgenes.com/specialty_oligonucleotide_synthesis.php

In fact you can purchase chemically modified 100mer sgRNA for \$299 or £226 from the first company on the list.

3. There is no characterization of your "Click" sgRNA 100mer products. You show HPLC chromatograms and mass spectra of 5'-N3 converted 79-mers. Where are the LC/MS characterizations of your sgRNAs? You make an argument that this is an alternative to making 100mer RNAs but I have no idea what your 100mer products look like in comparison to a routine synthesis of a 100mer RNA. I have included LC/MS chromatograms of HPLC purified 100mer sgRNA that are routinely generated in my laboratory; these are not the highest purity, but typical. We make them in a Dr. Oligo 96-well synthesizer and the typical crude yield of full-length product for a 100mer is 55% by ion-pair chromatography. After collecting fractions and evaluating them by LC/MS we typically isolate 1.5 mgs of full-length product from a 0.8 micromole synthesis that is 85 to 90% pure based upon the MS data; your analytical data should as good or better than what is shown to justify your claims in this manuscript. All of the steps to make the oligos shown are automated.

There are 3 to 5 published methods that allow for the chemical synthesis of RNA up to 120 nucleotides in length: *J. Am. Chem. Soc.*, 120, 11820-11821 (1998); *Chem. Eur. J.*, 14, 9135-9138 (2008); *J. Am. Chem. Soc.*, 133 (30), pp 11540-11556 (2011); *Curr. Pro. in Nuc. Acid Chem.*; Unit 3.20, (2011). I have additionally seen Trilink Biotechnologies claim that they use the chemistry described in *Helv. Chim. Acta*, 84, 3773-3795 (2001) to make 120mer RNAs.

My biggest concern with your approach is the use of the copper(I) catalyst. It is incredibly difficult to remove copper ions from long RNA and DNA (polyanions). From the experience in my laboratory the attempts to use CuAAC coupling on DNA or RNA makes the oligos very difficult to purify and analyze by LC/MS. If I recommended publication, I would want to see something like quantitative ICPMS data on the residual copper content; especially with your previous assertion of the use of these materials for therapeutics.

4. Once again, I do not recommend this manuscript for publication. I do not believe that there is any advantage of using CuAAC ligation to produce 100mer RNAs regardless of whether they are guide RNAs, especially when I can't find any significant chemical analysis of the resulting products to demonstrate an advantage over straight chemical synthesis. If the authors had shown that they could stitch together a functional 1K messenger RNA from 10 "routine-to-synthesize" 100mer RNAs using CuAAC ligation, that would be something I would clearly support to a broad readership. The idea of using CuAAC ligation to put shorter oligonucleotides together and evaluating modes and positions where by they are tolerated in vivo was a good idea. I just have yet to have seen that promise result in a compelling biological application, and 100mer RNA are just way too easy to make to justify the additional chemistry required by CuAAC ligation.

Reviewer #3 (Remarks to the Author):

The revised manuscript has addressed all of our concerns.

We thank the three reviewers for their kind words and feedback on our manuscript. Below is our detailed response to Reviewer 2.

Reviewer 2. “Click and play CRISPR sgRNA” is not the title of a scientific manuscript.

We have revised the title to:

An Artificial Triazole Backbone Linkage Provides a Split-and-Click Strategy to Bioactive Chemically Modified CRISPR sgRNA

Reviewer 2. There is no characterization of your “Click” sgRNA 100mer products. You show HPLC chromatograms and mass spectra of 5'-N3 converted 79-mers. Where are the LC/MS characterizations of your sgRNAs? My biggest concern with your approach is the use of the copper(I) catalyst. **Editor:** Reviewer 1 suggested that you could provide at least a representative MS (not necessarily LC - since you look at the clicked product on a gel) of a long clicked oligo, this should satisfactorily address the concern about copper contamination.

Reviewer 1 requested a representative mass spectrum of a long clicked oligo. We now provide both HPLC chromatograms and mass spectra for each type of modified clicked ~99-mer sgRNA (all RNA, DNA or OME; the final design used in cells), and have reanalysed the 79-mer oligo data to improve the signal quality. These results are in the revised Supplementary Figure 1 shown below. Furthermore, we have carried out MS characterisation of all twenty-four ~99-mer clicked sgRNA constructs for which the observed and calculated mass values all match (Supplementary Table 1 and 3).

Regarding copper contamination of the click products – this would be seen as a +63 adduct by mass spectrometry but no peak corresponding to such adducts was present.

Supplementary Figure 1 | Representative HPLC chromatograms and mass spectra of 3'-propargyl 21-mer, 5'-N₃ converted 79-mers and click sgRNA-NH₂ constructs for in cell EMX1 target. CR-EMX1 (21-mer) [M]⁻ expected mass: 7100; found mass: 7102. TR1 (all RNA 79-mer) [M]⁻ expected mass: 25436; found mass: 25440. TR2 (RNA-DNA 79-mer) [M]⁻ expected mass: 25018; found mass: 25020. TR3 (RNA-OMe 79-mer) [M]⁻ expected mass: 26295; found mass: 26295. Click sgRNA-NH₂ (EMX1) [M]⁻ expected mass: 32536; found mass: 32540. Click sgRNA-DNA-NH₂ (EMX1) [M]⁻ expected mass: 32118; found mass: 32121. Click sgRNA-OMe-NH₂ (EMX1) [M]⁻ expected mass: 33395; found mass: 33397. Note that sodium ion adducts were also detected. The sequences corresponding to the oligonucleotide codes CR-EMX1, TR1, TR2 and TR3 can be found in Supplementary Table 5, and those corresponding to click sgRNA-NH₂ (EMX1), click sgRNA-DNA-NH₂ (EMX1) and Click sgRNA-OMe-NH₂ (EMX1) in Supplementary Table 3.

Editor. Reviewer 1 suggests your method is a screening approach rather than an alternative to synthesis of specific long sgRNAs. This needs discussion at least.

We discussed this briefly at in the conclusion previously and have now expanded upon it as follows –

The successful use of the bio-compatible triazole linkage, Tz2, allows radical fragmentation of the sgRNA. For example, the sgRNA could be split into multiple parts and combinatorially reassembled to access diverse sgRNA libraries containing chemical²⁴ or sequence modifications. Notably, the greatest strength of our split-and-click approach lies in screening numerous individual sgRNAs for function rather than preparing a single sgRNA. As the number of sgRNAs needed increases, the costs and time associated with repeated full length sgRNA synthesis become greater and the merits of our approach become more evident. Moreover, 'clicked' sgRNAs provide easier access to bespoke pools of modified sgRNAs (*c.f.* full length sgRNA synthesis). Libraries of ~20-mer RNAs could be archived and custom libraries generated economically on demand with short turnaround times between sgRNA design and application.

Editorial note: To resolve any outstanding concerns, the editor asked Reviewer #1 to assess the experimental data added to the manuscript at the final round of review, in lieu of Reviewer #2. Their comments are displayed below.

REVIEWERS' COMMENTS:

Reviewer #1 (Remarks to the Author):

Glad to see the revised version that takes into account all the reviewers' comments, and the responses to reviewers.

The idea to split and click the sgRNA to enable higher throughput in CRISPR screens is excellent. I do recommend this for publication in Nature Communications.

My only comment specific to this version of the manuscript - I do find the revised title quite verbose. The initial phrase seems unnecessary. Here are a couple of *suggestions*:

A Split-and-Click Strategy to Bioactive Chemically Modified CRISPR sgRNA

or

Click and play sgRNA towards higher throughput CRISPR screens

Ultimately as far as the title is concerned, should the rest of the manuscript be acceptable to the editor, I leave it to the discretion of the authors and editor to decide on a suitable title